# The Study of Exotic and Invasive Plant Species in Gullele Botanic Garden, Addis Ababa, Ethiopia

Mehari Girmay [1,*], Kflay Gebrehiwot [2,3], Ergua Atinafe [1], Yared Tareke [4] and Birhanu Belay [1]

1    Plant Research Directorate, Gullele Botanic Garden, Addis Ababa P.O. Box 153/1029, Ethiopia;
     ergoye21@gmail.com (E.A.); birhanubelay79@gmail.com (B.B.)
2    Department of Biology, Samara University, Semera P.O. Box 132, Ethiopia; kflaygebre@su.edu.et
3    Applied Behavioural Ecology and Ecosystem Research Unit, School of Ecological and Human Sustainability,
     University of South Africa, Pretoria P.O. Box 1710, South Africa
4    Horticulture and Development Directorate, Gullele Botanic Garden, Addis Ababa P.O. Box 153/1029, Ethiopia;
     yaredgbg@gmail.com
*    Correspondence: mehari.girmay@aau.edu.et or meharigrm@gmail.com

**Abstract:** The Gullele Botanic Garden was established to preserve and safeguard indigenous, rare, endemic, endangered, and economically important plant species. The objective of this study was to identify and map the exotic, invasive, and potentially invasive plant species that are present in the garden's various land use types, such as natural vegetation, plantations, roadsides and garden edges. The research involved laying plots at different distances in each land use type and collecting vegetation data with geo-location information. Sorensen's similarity index was used to measure the floristic similarity between the sampled land use types. Data on species density and abundance were analyzed using the corresponding formula. The Shannon–Wiener diversity index and evenness were used to compute the diversity of the species in each land use type using R packages. ArcGIS version 10.5 was used to track the geographical distribution and map the exotic, invasive, and potentially invasive species that exist in all land use types of the garden. A total of 80 plant species belonging to 70 genera in 44 families were recorded in the garden. *Asteraceae*, *Myrtaceae*, and *Fabaceae* comprised the highest number of species. *Acacia decurrens*, *Acacia melanoxylon*, *Cuscuta campestris*, *Galinsoga parviflora*, *Nerium oleander*, and *Cyathula uncinulata* were the most prevalent invasive and potentially invasive species. The study found that the roadside and garden edge land use types had the most diverse exotic and invasive plants. The total density of exotic species was 2.36 plants/m$^2$. The potential possibility of these plants in displacing the native plant species is quite high unless the introduction of exotic plant species is inspected and appropriate management strategies for invasive species are put in place.

**Keywords:** ex situ conservation; exotic species; invasive species; native species; non-native




## 1. Introduction

An exotic species or alien species is any non-native plant, animal, or other organism introduced into a place that was never part of its natural range. The presence of exotic taxa in specific places is mostly the result of either anthropogenic activities or natural processes [1,2]. For example, certain exotic species require human involvement or cultivation in order to be introduced into a certain area. On the other hand, some species may naturalize by sustaining their own populations in the absence of human interference [3]. The manner of occurrence and introduction may either be intentional for certain economic and other values, or species may be introduced accidentally or unintentionally with other vectors [4].

The planting and rehabilitation efforts of parks, gardens and several agroecosystems depend on non-native (exotic) plant species. However, introducing these species leads to a serious threat to plants or to the entire biodiversity in an ecosystem from their potential

invasion and unnecessarily proliferation out of their desirable range [4,5]. This has the effect of reducing or removing the advantages of agroforestry for biodiversity [5]. On the other hand, several researchers, including [1,2,6], have suggested that not all exotic species in a particular ecosystem have an adverse impact. They underscored that the invasiveness feature should be researched before they are introduced to a given habitat.

Different definitions have been given for the term invasive alien species. It has been described as a non-indigenous species that spreads and becomes abundant outside the normal range of the native plant population after being introduced to the given habitat [7]. Notably, some native species may have invasive features. Other studies have described invasive species as biological invaders, mostly transported inadvertently or intentionally by man, whereupon they colonize and spread into other areas, sometimes far from their home territory [8,9]. According to Krishnan and Novy [5] and Richardson et al. [3], invasive alien species pose a threat to ecosystems, habitats, or species by either (1) eluding human control, (2) extending beyond their intended physical boundaries, or (3) remaining under human control but causing harm to native ecosystems. Plant invasions have been recognized as one of the most serious global phenomena impacting the structure, composition, and function of natural and semi-natural ecosystems. Due to their rapid growth and management difficulties, they outpace the local biota in terms of habitat occupancy and resource exploitation [10]. This damage is aggravated by climate change, pollution, habitat loss, and human-induced disturbance [11]. The threat is extensive, particularly in areas where plant communities are disturbed [12,13]. Currently, the issue is only becoming worse, at a significant cost to society, the environment, and the economy everywhere in the world, particularly in the tropics [10].

Even though Ethiopia is endowed with plant genetic resources linked to geographic diversity, macro and microclimatic variability, and existing abundant species, unknown numbers of exotic species have been introduced over the years for different purposes [14,15]. Of these, there are invasive and potentially invasive species [16]. The lack of consideration for the source of materials used in restoration and planting works in Ethiopia is likely the reason for the majority of these works being executed without regard for the biodiversity in forest regions, national parks, and damaged ecosystems [17]. In Ethiopia, studies both on invasive and alien plant species are scant, even though numerous invasive species are spreading throughout the country [16]. A certain degree of attention has been devoted to invasive taxa over the past ten years. This will play a major role in the conservation of the entire vegetative ecosystem as well as indigenous plant species [17].

Currently, botanic gardens are a good strategy for the ex situ conservation of plant species by taking their nativity and threating status [5,18]. Botanic gardens in Ethiopia are being used to conserve indigenous, endangered, endemic, and economically important plant species as well as preserve the country's rarest species. The Gullele Botanic Garden (GBG) is the foremost botanic garden in Ethiopia with the main objectives of fulfilling plant conservation, research, education, and eco-tourism. The conservation work prioritizes indigenous, threatened, endemic, endangered, and economically important plants as well as rare species found in the country. The organization has developed different infrastructures that enable it to operate fairly within its vision and mission. The garden conducts various in situ and ex situ conservation techniques through the development of a thematic garden and an evolution garden with collections of plants and seeds from different agro-ecologies of the country [5]. Those species from distinctive areas are acclimatized in the existing greenhouse. Currently, more than 1300 species have been introduced to the naturally occurring species in the garden using collection and in situ management approaches, which are accompanied by 2200 specimen deposits.

The Gullele Botanic Garden plays a significant role in the preservation and maintenance of the country's plant genetic resources. It is currently working hard and succeeding in a variety of duties that help it realize its objective. One of the organization's main goals is to conduct research and conserve the indigenous flora. However, either intentionally or accidentally, some exotic species (including invasive species) have existed in the gar-

den. The types, distribution, abundance and magnitude of invasion of these exotic plant species inside the botanic garden are not known or studied yet. Hence, this study will have significant importance to make apposite decisions and mitigate the adverse effects of invasive and potential invasive species on the indigenous species. Hence, this study was conducted to explore the existing exotic, invasive and potentially invasive species and map their spatial distribution in the garden for their future management.

## 2. Materials and Methods

### 2.1. Study Area Description

The Gullele Botanic Garden is situated between 2540 and 3000 m above sea level, particularly northwest of Addis Ababa, with coordinates between 9°1′30″ and 9°5′35″ N and 38°41′30″ and 38°44′20″ E (Figure 1). It is a section of Ethiopia's central plateau and covers 705 hectares. Both hot and cold weather simultaneously can occur in the area. February is the warmest month (20.7 °C) followed by March and May with 20.2 and 20 °C, respectively. December has the lowest average temperature (7.5 °C). The dry season lasts from March to May, and there is an average of 1215.4 mm of precipitation [19,20]. Dry afro-montane dominates the vegetation type in the study area with a smaller amount of afro-alpine in the elevated areas of the garden. *Juniperus procera* is the most dominant plant species in the garden. Next to the *Juniperus procera*, a variety of herbaceous species are co-dominant alongside woody species such as *Rosa abyssinica*, *Olinia rochetiana*, *Jasminum abyssinicum*, *Myrsine africana*, *Sideroxylon oxyacanthum*, *Maesa lanceolata*, *Maytenus species*, *Jasminum stans*, and *Vernonia Leopoldi*. In the elevated areas, various *Helichrysum* species and *Erica arborea* were frequently revealed. Previously, the garden was covered with *Eucalyptus* species, which are now being removed and managed in order to promote the growth of the prioritized native species. Silicic rocks predominate near Entoto, where the Gullele Botanic Garden is located [21]. This rock structure is named after a 21.5 million-year-old heal that borders the northern section of Addis Ababa. Trachyte and Rhyolite best characterize this type of rock. The garden has conducted research and development activities in addition to a variety of conservation efforts for threatened and endangered species collected from various parts of the country since it was recognized as a botanic garden by both national and international organizations.

### 2.2. Data Collection

Vegetation Data Collection

Vegetation data were gathered from a total of 64 plots that were purposely laid in the selected land use types—namely, natural vegetation, plantation, roadside and edges of the garden—following the sampling approach as described by [22,23]. Subsequently, the garden was stratified into three different land use types before ecological data collecting began.

1st strata: This includes the road and roadsides of the garden, the edge or the delimitation of the garden which encroaches to the agriculture, residences and external roads. In this case, vegetation data were gathered following the footpath transects and edges.

2nd strata: The vegetation in the garden that were planted at various times as well as thematic gardens were included in this land use type.

3rd strata: This included the entire Gulllele Botanic Garden's natural vegetation.

A 5 m by 5 m plot was purposely laid out in each land use types with a flexible distance range where exotic and invasive plants were seen in profusion for the first and second stratification. For the 3rd strata (natural forest), four transects with comparable plot sizes to the others were laid along an elevation gradient. The distance between each plot and transects were determined based on the targeted species abundance. This enables capturing fairly the targeted species in the garden. The number of plots was distributed in accordance with the species' abundance in each land use types seen during the reconnaissance survey. Accordingly, of the total number plots laid, 29 of them were taken from the roadside and edges of the garden, while the remaining 19 and 16 were taken from the plantation and natural forest, respectively.

Geospatial data—particularly, altitude, latitude, and longitude—were recorded from each plot. Subsequently, every plot was subject to GPS tracking, which is used to map out the locations of the plants.

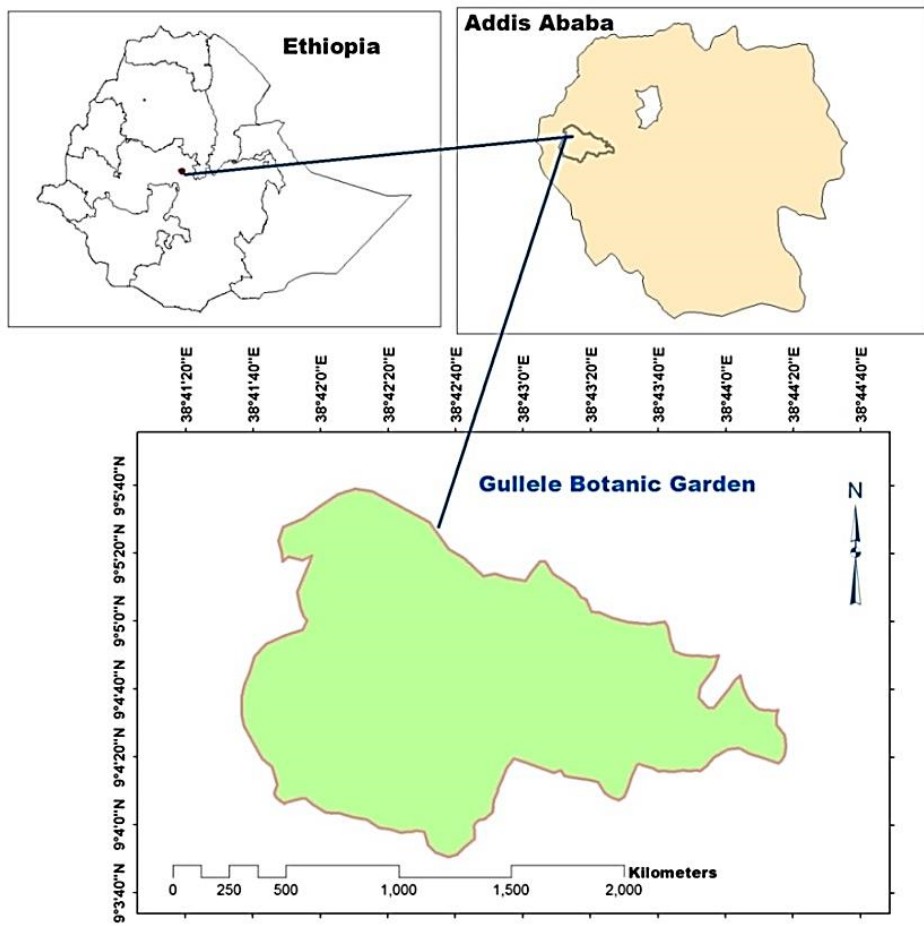

**Figure 1.** Map of the study area (Gullele Botanic Garden).

*2.3. Data Analysis*

2.3.1. Vegetation Data Analysis

The data relating to the abundance, density and diversity of the sampled species were analyzed using corresponding techniques and formulas. Species abundance was computed by counting the number of individuals in the sample plots, whereas density was computed by counting individual species per unit sample area using the following formula (Equation (1)):

$$\text{Density} = \frac{\text{Nnumber sampled species}}{\text{sample area}} \tag{1}$$

Species diversity in the three land use types was determined using the Shannon–Wiener diversity index and [23].

2.3.2. Floristic Similarity Analysis between Land Use Types

To examine the species composition similarity between the sampled land use types of the garden, Sorensen's similarity coefficient was employed. The value of the similarity coefficient ranges from 0 (total dissimilarity) to 1 (total similarity). This method is preferred because it gives weight to species that are common to the sample plots rather than to those

that only occur in either sample plot [13]. Sorensen's similarity index is calculated from the following equation (Equation (2)):

$$Ss = \frac{2a}{2a + b + c} \tag{2}$$

where *Ss* = Sorensen's similarity coefficient, *a* = the number of species common to both plant community types, *b* = the number of species present in one of the plant community types to be compared and *c* = the number of species present in the other plant community type.

### 2.3.3. Spatial Data Analysis

ArcGIS version 10.5 was used to map the distribution of exotic, invasive and potentially invasive species by taking geographical coordinate points from each sample plot. Following that, maps were created that show the locations of the 64 spatial coordinate points containing both invasive and exotic species.

## 3. Result
### 3.1. Exotic, Invasive and Potentially Invasive Plant Species in Gullele Botanic Garden

A total of 80 exotic, invasive and potentially invasive plant species (Appendix A) belonging to 70 genera in 44 families were recorded and identified from 64 plots in Gullele Botanic Garden. The highest number of species was recorded for the families Asteraceae (7 species, 8.8%), Myrtaceae (6 species, 7.5%), and Fabaceae (5 species, 6.3%) followed by families Verbenaceae, Agavaceae (4 species 5% each), Asparagaceae, Euphorbiaceae, Lythraceae and Poaceae (3 species, 3.8% each). The remaining families comprise ≤2 species each (Figure 2).

Regarding the habit of the exotic, invasive and potentially invasive species (Figure 3), herbs were predominant (33 species, 41%) followed by shrubs (25 species, 31%) and trees (22 species, 28%).

### 3.2. Exotic, Invasive and Potentially Invasive Species Abundance and Density in Different Land Use Types

The sampled three land use types were combined to calculate the abundance and density of exotic species. The abundance of the exotic, invasive and potential invasive species in the total sampled area of 1600 m$^2$ of the garden was 1458. Of these, 1045 species were recorded in the 725 m$^2$ sampled area laid for the roadside and edges. The remaining 285 species and 128 species were recorded in the sampled plots of plantations and natural forest, respectively (Table 1).

**Table 1.** Exotic and invasive/potential invasive species abundance and density in different land use types of GBG.

| Land Use Types | Abundance | Sampled Area (m$^2$) | Density (Species/m$^2$) |
|---|---|---|---|
| Natural forest | 128 | 400 | 0.32 |
| Roadside and edges | 1045 | 725 | 1.44 |
| Plantation | 285 | 475 | 0.6 |
| Total | 1458 | 1600 | 2.36 |

In terms of species density, roadside and garden edges had the highest density of species (1.44 species/m$^2$), while the natural forest had the lowest density of exotic, invasive and potentially invasive species (0.32 m$^2$).

**Number of species**

Asteraceae
Myrtaceae
Fabaceae
Verbenaceae
Poaceae
Lythraceae
Euphorbiaceae
Agavaceae
Rosaceae
Melastomataceae
Geraniaceae
Commelinaceae
Bignoniaceae
Asparagaceae
Apocynaceae
Anacardiaceae
Aizoaceae
Lauraceae
Vitaceae
Solanaceae
Scrophulariaceae
Rutaceae
Proteaceae
Pinaceae
Papaveraceae
Orobanchaceae
Oleaceae
Nyctaginaceae
Moringaceae
Moraceae
Meliaceae
Malvaceae
Hydrangeaceae
Cuscutaceae
Cupressaceae
Crassulaceae
Celastraceae
Caryophyllaceae
Caricaceae
Cactaceae
Arecaceae
Araucariaceae
Amaranthaceae
Agapanthaceae

0     1     2     3     4     5     6     7     8

**Figure 2.** List of families and their corresponding number of species in Gullele Botanic Garden.

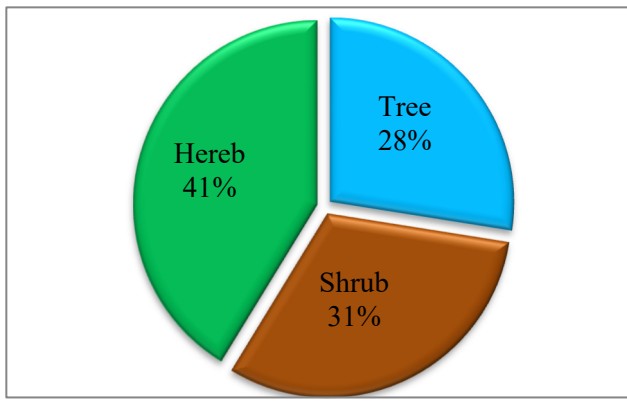

**Figure 3.** Percentage distribution of exotic species in GBG.

### 3.3. Exotic, Invasive and Potentially Invasive Species Diversity in Different Land Use Types

The Shannon–Wiener diversity index (Table 2) showed that the roadside and garden edge land use types had the highest diversity (3.09) of exotic and invasive plant species. Plantations (0.63) and natural forests (0.06), however, scored the lowest for Shannon–Wiener diversity. Similarly, samples collected from roadside and garden edges had the highest species evenness (0.82), whereas the samples taken from natural forests and plantations had low species evenness with values of 0.29 and 0.05, respectively.

**Table 2.** Shannon–Wiener diversity and evenness indices of exotic, invasive and potentially invasive species in various land use types of Gullelle Botanic Garden.

| Community | Shannon–Wiener Diversity Index (H′) | Shannon Evenness (J′) |
|---|---|---|
| Natural forest | 0.06 | 0.05 |
| Roadside and edges of the garden | 3.09 | 0.82 |
| Plantation | 0.63 | 0.29 |

### 3.4. Floristic Similarity Analysis between Land Use Types

Sorensen's similarity coefficient (Table 3) was used to compare the floristic composition similarities between the sampled land use types. The roadside and edges of the garden and plantation land use types had the highest similarity coefficient, while the least similarity was found between plantations and natural forest.

**Table 3.** Pair-wise comparison of Sorensen's similarity coefficient between the sampled land use types in Gullele Botanic Garden.

| Land Use Type | Natural Forest | Roadside and Edges of the Garden | Plantation |
|---|---|---|---|
| Natural forest | 1 | 0.22 | 0.15 |
| Roadside and edges of the garden | - | 1 | 0.40 |
| Plantations | - | - | 1 |

### 3.5. Invasive species in Gullele Botanic Garden

The 15 species recorded in Gullele Botanic Garden (Table 4) were invasive or have a potential invasive trait. Of these shrubs, herbs and trees were represented by six, six, and three species, respectively. Of these, 13 (86.7%) of them were found in the roadside and edges land use type. With the exception of two species, the remaining land use types shared different species with the roadside and edge land use types.

**Table 4.** List of invasive and potentially invasive species, habit (T = tree, S = shrub, H = Herb), land use types (LUT: land use type, NF = natural forest, RE = roadside and edge of the garden, PL = plantations).

| No. | Species Name | Family | Habit | LUT Found |
|---|---|---|---|---|
| 1 | *Acacia decurrens* Willd. | Fabaceae | T | NF, RE, PL |
| 2 | *Acacia mearnsii* De Wild. | Fabaceae | T | RE |
| 3 | *Acacia melanoxylon* R. Br. | Fabaceae | T | NF, RE, PL |
| 4 | *Acacia saligna* (Labill.) Wendl. | Fabaceae | S | NF, RE |
| 5 | *Argemone mexicana* L. | Papaveraceae | H | RE |
| 6 | *Cuscuta campestris* Yuncker | Cuscutaceae | H | NF, RE |
| 7 | *Cyathula uncinulata* (Schrad.) Schinz | Commelinaceae | H | NF, RE |
| 8 | *Galinsoga parviflora* Cav. | Asteraceae | H | PL |
| 9 | *Lantana camara* L. | Verbanaceae | S | NF, RE |

**Table 4.** *Cont.*

| No. | Species Name | Family | Habit | LUT Found |
|---|---|---|---|---|
| 10 | *Nerium oleander* L. | Apocynaceae | S | RE, PL |
| 11 | *Nicotiana glauca* Graham | Solanaceae | S | RE |
| 12 | *Psidium guajava* L. | Myrtaceae | S | PL |
| 13 | *Ricinus communis* L. | Euphorbiaceae | H | RE, PL |
| 14 | *Senna didymobotrya* (Fresen.) Irwin & Barneby | Fabaceae | S | RE |
| 15 | *Striga gesnerioides* (Willd.) Vatke | Scrophulariaceae | H | RE, PL |

Invasive Species Distribution and Land Use Types

The distribution of the invasive species in the sampled land use types of the garden is demonstrated in Figure 4. The majority of the studied land use types share various invasive species. However, some species, such as *Nicotiana glauca*, *Argemone mexicana*, and *Acacia mearnsii*, were restricted to the roadside and garden edge, while *Psidium guajava* and *Galinsoga parviflora* were restricted to the plantation. No specific species was restricted to the natural forest.

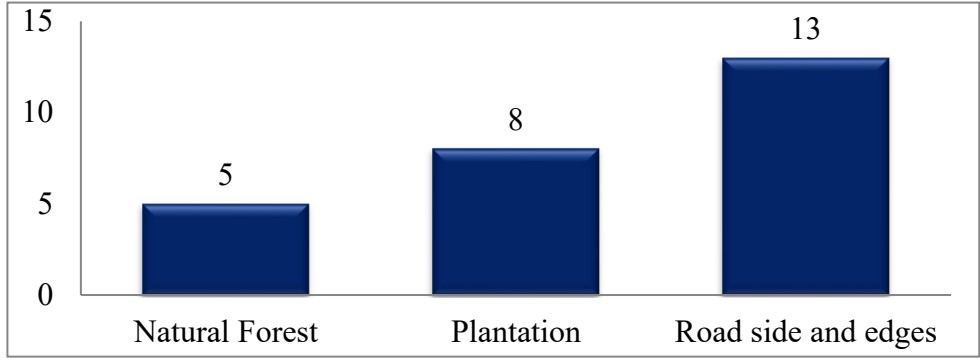

**Figure 4.** Distribution of invasive species in Gullele Botanic Garden.

The identified 15 invasive and potentially invasive species were found in 33 sample plots of the garden (Figure 5). Even though the majority of invasive plant species were found in the roadsides and garden edges, some species, such as *Acacia decurrens*, *Acacia melanoxylon*, *Cuscuta campestris*, *Cyathula uncinulata*, *Galinsoga parviflora*, *Nerium oleander*, *Senna didymobotrya* and *Striga gesnerioides*, were also widespread in plantations and natural forest.

*3.6. Spatial Distribution of Exotic Species in the Garden*

As indicated in the map (Figure 6), the ratio of the spatial distribution of the exotic species varies from one land use type to another. In every sampled plot across all land use types, there were more than two exotic species. The roadside and garden edge type possesses more than 76% of all exotic plants found in the garden. Species such as *Acacia decurrens*, *Acacia melanoxylon*, *Agave species*, *Callistemon citrinus*, *Grevillea robusta*, *Pinus patula*, *Duranta erecta*, *Phalaris arundinacea*, *Lavandula angustifolia*, *Tradescantia pallida*, *Bougainvillaea glabra*, *Pelargonium zonale*, *Cuphea* species, and *Eucalyptus* species that were planted at different times were observed in the garden's degraded area as well as in the existing themes. Of the exotic species collected from the natural forest of the garden, species including *Acacia decurrens*, *Acacia melanoxylon*, *Cupressus lusitanica*, *Osteospermum fruticosum*, *Melaleuca alternifolia* and *Eucalyptus* species were recorded.

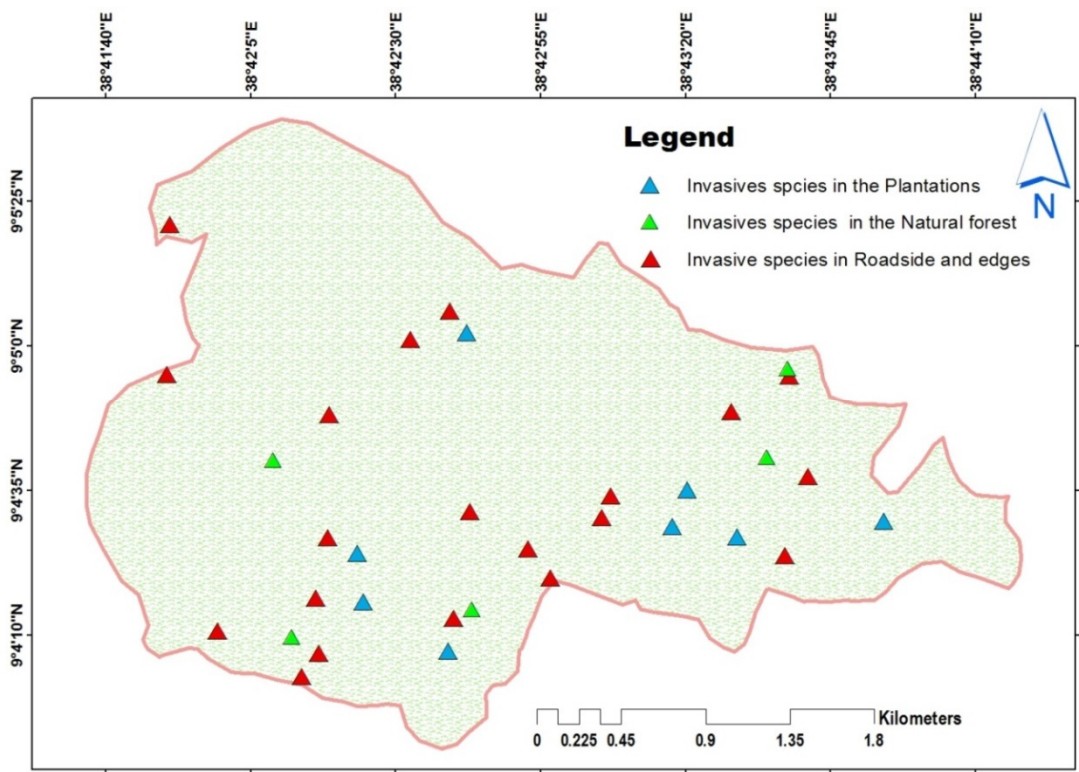

**Figure 5.** A spatial distribution map that shows the distribution of the invasive and potential invasive species in Gullele Botanic Garden.

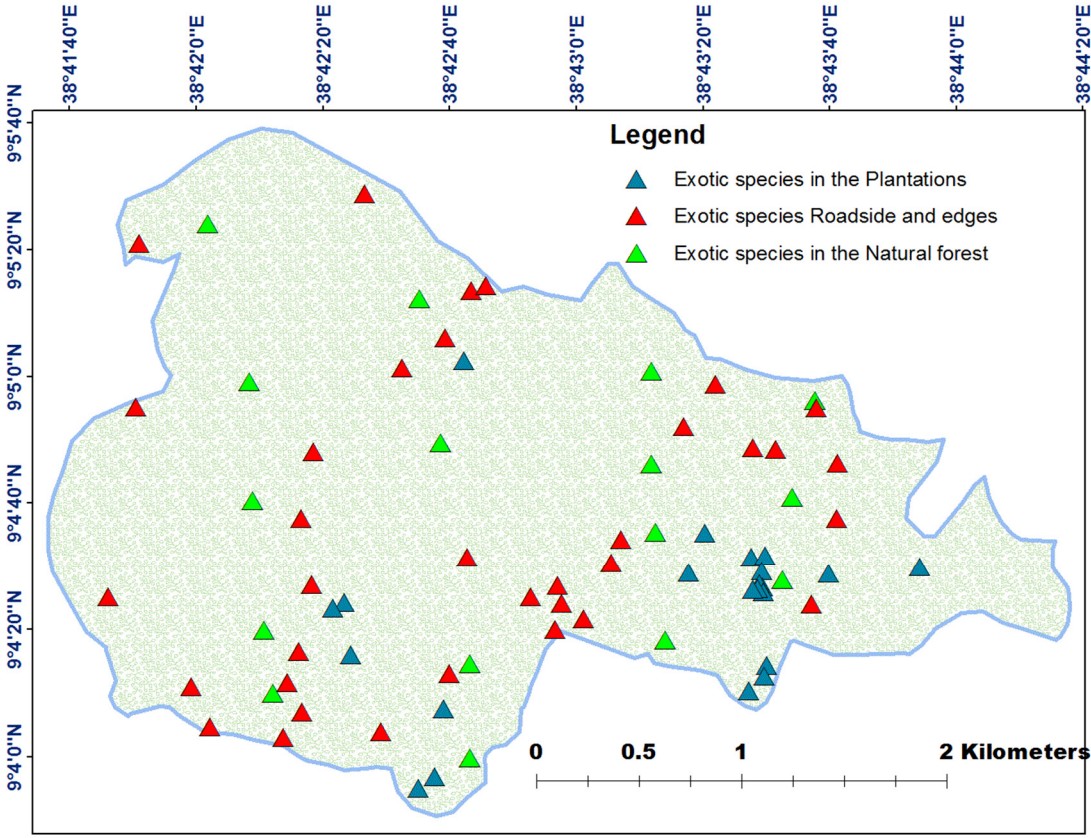

**Figure 6.** Spatial distribution exotic species (invasive and non-invasive species) in the garden.

## 4. Discussion

### 4.1. Exotic Plant Species in Gullele Botanic Garden

The presence of 80 non-native plant species in the garden suggests that they were intentionally or accidentally introduced at various times and might potentially spread throughout the garden. Studies conducted by Thomas et al. [24] and Vukeya et al. [25] have shown that the native species in many botanic gardens elsewhere in the world are at risk and the huge impact. This might be associated with their reliability on ex situ conservation through introducing uninspected exotic species. A similar situation was observed in the Gullele Botanic Garden. The abundance of exotic plants in the garden may be a result of their multitude of uses and ecological adaptability features [26].

The families Asteraceae, Myrtaceae and Fabaceae had the highest number of species, which is consistent with their status as species-rich families in the Flora of Ethiopia and Eritrea [27] as well as the East Africa region [16]. However, the presence of exotic species that belong to the families Asparagaceae and Lythraceae may be due to their desirability for horticultural purposes in the garden. Numerous studies conducted around the world have indicated that the introduction of ornamental plants is likely to increase the richness of exotic species [28,29].

According to a study on invasive and potential invasive species in Ethiopia [15] as well as in East Africa [30], about 15 of the plant species identified in the Gullele Botanic Garden have an invasive characteristic. The introduction of these species to the garden could occur either deliberately, by planting for their beneficial qualities, or unintentionally through imported seeds, vehicles, or other vectors and pathways. The abundance of herbaceous invasive species in the garden may be related to the species' ability to spread quickly through the use of seeds by various agents, including wind, birds, and humans and their suitability for the garden's ecosystem. Woody species were also abundant, which might be associated with the desirability for restoration of the degraded area. A similar result was also reported by Mokotjomela et al. [25].

### 4.2. Species Abundance, Density and Diversity in the Sampled Land Use Types

High exotic, invasive and potentially invasive species richness, density and diversity in the roadside and edge of the garden might be a result of whether the land use type is easily accessible by human and livestock. In Ethiopia, it is reported that about 80% of the invasive and potentially invasive alien species affected the roadside ecosystem and agricultural areas which are exposed to anthropogenic interventions [15,31]. Furthermore, most plantations are located in and around the bare, degraded areas that edge the garden and the paths used by humans. In contrast to this, the least exotic, invasive and potentially invasive species abundance, density and diversity was recorded in the natural forest. The least exposure to humans and livestock in the natural forest might contribute to having a least exotic, invasive and potentially invasive species [2,32,33].

The fact that invasive species exist across all types of land use suggests that the species has an aggressive mode of dispersion [33,34]. Numerous invasive plants were found even inside the natural forest; they may have spread naturally or as a result of unintentional human intrusions from the garden's edge and roadside. However, species like *Sidium guajava* and *Ricinus communis* were deliberately introduced for their economic benefits either through plantations or delivered in either as part of plantations or in combination with other species. According to a study by Haber [11], it will be expensive to control the spread of invasive species once they reach their climax distribution if they are not managed in the early stages of dispersion.

Regarding the distribution of invasive species in relation to land use types, the roadsides and edges of the garden were more dominant locations than the plantations and natural forest. Even within the plots taken from the edge and roadside land use type itself, high numbers of invasive species were recorded at the lower side of the garden. This is associated with the sloping terrain of the garden, allows the seeds of the species to easily migrate to lower edges of the garden during the rainy season. This is consistent with

research by Oh et al. [35] that found that water flows, in addition to other anthropogenic variables, had a positive effect on the spread of invasive species. Another study conducted by Richardson et al. [34] revealed that roadsides, river basins, and horticultural gardens coupled with degradations, grazing and deforestation [30,36] may be the primary sources of invasive species dispersal in East Africa specifically in Ethiopia [30,37]. Some of the potentially invasive species such as *Acacia decurrens*, *Cyathula uncinulata*, *Acacia melanoxylon* and *Eucalyptus* species were aggressively dominating the indigenous species. This might have a serious ecological adverse impact for the future. A study conducted by [30,37] showed that a large number of species in Africa are introduced to native countries from different aspects of the world either deliberately or unintentionally from their natural habitats through human (e.g., agro-forestry, horticulture, forestry, and animal husbandry purposes) or natural (e.g., winds, birds, animals, water). The IUCN [28] report showed that few of these introduced species to a given ecosystem become problematic and have the potential to invade the native species within the near coming future.

### 4.3. Spatial Distribution of Exotic and Invasive Species in the Garden

Exotic, invasive and potentially and invasive species were prevalent along human pathways and in places that encroached on residential and agricultural areas. This is a result of the native species in these areas being exposed to logging and being replaced by other exotic species [28]. Horticulturalists and gardeners in many tropical nations use non-native plants without taking into account the detrimental ecological impacts [11,29,33]. Subsequently, many exotic species aggressively cover a vast area by replacing native species [29]. These species can be introduced with crops and other mobile objectives [11,25]. Although most of the exotic species showed the feature of dominance over the other native species in the garden, the high spatial distributions and occupancies were revealed by the invasive species such as *Acacia decurrens*, *Acacia melanoxylon*, *Acacia saligna*, *Cyathula uncinulata*, *Senna didymobotrya* and *Nerium oleander*. This might be associated with the nature of the aggressiveness and dominating future of invasive species by exploiting the macro- and micronutrients in the given ecosystem [1,6]. However, the least spatial prevalence of the exotic species in the natural forest suggests that there might be a sort of attention on species invasiveness and halted human accessibilities [1,29,31].

### 5. Conclusions

In the Gullele Botanic Garden, 80 exotic, invasive and potentially invasive species were recorded. These plant species were introduced in recent times and are associated with plantations, unintentionally spreading with other vectors. Currently, a high population size was recorded in and around the roadsides and vicinities to the residences. The 15 invasive species in the garden would have a serious adverse impact on the native plant species in particular and the forest ecosystem of the garden in general. However, the introduction, cultivation and conservation of these species is not compatible with the objective of the institution. Unless earlier management practices are implemented, they have a huge potential to dominate the native species of the garden. Therefore, since this study has identified the list of invasive and exotic species with their spatial distribution, prior management techniques, including the halting of deliberate or unintentional introduction of invasive species, removal of seedlings and seeds of the invasive species, periodical monitoring and giving plantation priority to the native plants or replacing the exotic/invasive species by native species should be implemented. The empirical data from this study will aid the decision-makers in taking proper management measures on the exotic, invasive and potentially invasive species in the Gullele Botanic Garden.

**Author Contributions:** Conceptualization, methodology, validation, M.G. and K.G.; software, M.G.; formal analysis, investigation, M.G.; resources, E.A. and Y.T.; data curation, M.G., E.A. and Y.T.; writing—original draft preparation, M.G.; writing—review and editing, M.G., K.G., E.A., Y.T. and B.B.; visualization, M.G.; supervision, B.B. All authors have read and agreed to the published version of the manuscript.

**Funding:** This research received no external funding.

**Institutional Review Board Statement:** Not applicable.

**Data Availability Statement:** Data are contained within the article.

**Acknowledgments:** The authors acknowledge the Gullele Botanic Garden and staffs who allowed this research and cooperate whilst they were requested a support.

**Conflicts of Interest:** The authors declare no conflict of interest.

## Appendix A. List of Exotic Species

**Table A1.** Habit: T = tree, S = shrub, H = herb, C = climber. Status invasiveness: I = invasive, PI = potentially invasive, NI = not invasive, * non-exotic species but included because of their invasive feature.

| No. | Species Name | Family | Habit | Status Invasiveness |
|---|---|---|---|---|
| 1 | *Acacia decurrens* Willd. | Fabaceae | T | PI |
| 2 | *Acacia mearnsii* De Wild. | Fabaceae | T | PI |
| 3 | *Acacia melanoxylon* R. Br. | Fabaceae | T | I |
| 4 | *Acacia saligna* (Labill.) Wendl. | Fabaceae | S | I |
| 5 | *Agapanthus africanus* T.Durand and Schinz | Agapanthaceae | H | NI |
| 6 | *Agave americana* L. | Agavaceae | S | NI |
| 7 | *Agave sisalana* Perro ex Eng. | Agavaceae | S | NI |
| 8 | *Aloysia triphylla* (L'Herit.) Britton | Verbenaceae | S | NI |
| 9 | *Aptenia cordifolia* (Tenore) V. Steenis | Aizoaceae | H | NI |
| 10 | *Araucaria heterophylla* (Salisb.) Franco | Araucariaceae | T | NI |
| 11 | *Argemone mexicana* L. | Papaveraceae | H | I |
| 12 | *Arundo donax* L. | Poaceae | H | NI |
| 13 | *Azadirachta indica* A. Juss. | Meliaceae | T | NI |
| 14 | *Bougainvillea glabra* | Nyctaginaceae | S/C | NI |
| 15 | *Callistemon citrinus* (Curtis) Skeels. | Myrtaceae | T | NI |
| 16 | *Callistephus chinensis* (L.) Nees | Asteraceae | H | NI |
| 17 | *Carica papaya* L. (Caricaceae). | Caricaceae | T | NI |
| 18 | *Carpobrotus edulis* (L.) L. Bolus | Aizoaceae | H | NI |
| 19 | *Centradenia floribunda* shawl | Melastomataceae | H | NI |
| 20 | *Chlorophytum comosum* (Thunb.) Jacques | Asparagaceae | H | NI |
| 21 | *Chrysanthemum leucanthemum* L. | Asteraceae | H | NI |
| 22 | *Citrus aurantiifolia* (Christm.) Swingle | Rutaceae | S | NI |
| 23 | *Cordyline australis* (G.Forst.) Endl. | Asparagaceae | T | NI |
| 24 | *Cordyline fruticosa* (L.) A.Chev. | Asparagaceae | H | NI |
| 25 | *Crassula ovata* (Miller) Druce | Crassulaceae | H | NI |
| 26 | *Cuphea hyssopifolia* Kunth | Lythraceae | H | NI |
| 27 | *Cuphea ignea* A.DC | Lythraceae | H | NI |
| 28 | *Cuphea micropetala* Kunth | Lythraceae | H | NI |
| 29 | *Cupressus lusitanica* Mill | Cupressaceae | T | NI |
| 30 | *Cuscuta campestris* Yuncker | Cuscutaceae | H | I |

**Table A1.** *Cont.*

| No. | Species Name | Family | Habit | Status Invasiveness |
|---|---|---|---|---|
| 31 | *Cyathula uncinulata* (Schrad.) Schinz * | Commelinaceae | H | I |
| 32 | *Cymbopogon citratus* (DC.) Stapf. | Poaceae | H | NI |
| 33 | *Dianthus barbatus* L. | Caryophyllaceae | H | NI |
| 34 | *Distictis buccinatoria* (DC.) A.H. | Bignoniaceae | HC | NI |
| 35 | *Duranta erecta* L. | Verbenaceae | S | NI |
| 36 | *Duranta repens* L. | Verbenaceae | S | NI |
| 37 | *Eucalyptus camaldulensis* Dehnh. | Myrtaceae | T | NI |
| 38 | *Eucalyptus citriodora* Hook. | Myrtaceae | T | NI |
| 39 | *Eucalyptus globulus* Labill. | Myrtaceae | T | NI |
| 40 | *Euonymus fortunei* (Turcz.) Hand.-Mazz. | Celastraceae | S | NI |
| 41 | *Euphorbia milii* Des Moulins | Euphorbiaceae | S | NI |
| 42 | *Ficus benjamina* Linn. | Moraceae | S | NI |
| 43 | *Galinsoga parviflora* Cav. | Asteraceae | H | I |
| 44 | *Grevillea robusta* R.Br. | Proteaceae | T | NI |
| 45 | *Hibiscus rosa-sinensis* L. | Malvaceae | S | NI |
| 46 | *Hydrangea macrophylla* (Thunb.) Ser. | Hydrangeaceae | H | NI |
| 47 | *Iresine herbstii* Lindl. | Amaranthaceae | H | NI |
| 48 | *Jacaranda mimosifolia* D.Don | Bignoniaceae | T | NI |
| 49 | *Jatropha curcas* L. | Euphorbiaceae | T | NI |
| 50 | *Lantana camara* L. | Verbenaceae | S | I |
| 51 | *Lavandula angustifolia* Mill. | Lauraceae | S | NI |
| 52 | *Ligustrum vulgare* L. | Oleaceae | S | NI |
| 53 | *Malus domestica* Borkh. | Rosaceae | S | NI |
| 54 | *Mangifera indica* L. | Anacardiaceae | T | NI |
| 55 | *Melaleuca alternifolia* Cheel. | Myrtaceae | T | NI |
| 56 | *Moringa oleifera* Lam. | Moringaceae | T | NI |
| 57 | *Nerium oleander* L. | Apocynaceae | S | I |
| 58 | *Nicotiana glauca* Graham | Solanaceae | S | I |
| 59 | *Opuntia ficus-indica* (L.) Miller. | Cactaceae | S | NI |
| 60 | *Orobanche minor* Smith | Orobanchaceae | H | NI |
| 61 | *Osteospermum fruticosum* (L.) Norl. | Asteraceae | H | NI |
| 62 | *Pelargonium asperum* Willd. | Geraniaceae | H | NI |
| 63 | *Pelargonium zonale* (L.) L'Hér. | Geraniaceae | H | NI |
| 64 | *Persea americana* Mill. | Lauraceae | T | NI |
| 65 | *Phalaris arundinacea* L. | Poaceae | H | NI |
| 66 | *Pinus patula* Schiede ex Schltdl. & Cham. | Pinaceae | T | NI |
| 67 | *Psidium guajava* L. | Myrtaceae | S | I |
| 68 | *Ricinus communis* L. * | Euphorbiaceae | H | I |
| 69 | *Rosa pendulina* L. | Rosaceae | S | NI |
| 70 | *Schinus molle* L. | Anacardiaceae | T | NI |

**Table A1.** *Cont.*

| No. | Species Name | Family | Habit | Status Invasiveness |
|-----|--------------|--------|-------|---------------------|
| 71 | *Senecio cineraria* | Asteraceae | H | NI |
| 72 | *Senna didymobotrya* (Fresen.) Irwin & Barneby * | Fabaceae | S | I |
| 73 | *Silybum marianum* (L.) Gaertn. | Asteraceae | H | NI |
| 74 | *Striga gesnerioides* (Willd.) Vatke * | Scrophulariaceae | H | I |
| 75 | *Tagetes minuta* L. | Asteraceae | H | NI |
| 76 | *Tibouchina urvilleana* (DC.) Cogn. | Melastomataceae | T | NI |
| 77 | *Tradescantia pallida* (Rose) Hunt | Commelinaceae | H | NI |
| 78 | *Vinca major* L. | Apocynaceae | HC | NI |
| 79 | *Vitis vinifera* L. | Vitaceae | WC | NI |
| 80 | *Washingtonia filifera* (Linden ex Andre) H. Wendl. | Arecaceae | S | NI |

Sources [10,29,35].

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
