# Peer review of "The Study of Exotic and Invasive Plant Species in Gullele Botanic Garden, Addis Ababa, Ethiopia"

_2673-5636, doi:10.3390/jzbg5010003_

Round 1

Reviewer 1 Report

Comments and Suggestions for Authors

Dear Editor in chief,

Journal of Zoological and Botanical Gardens (JZBG), MDPI

I would like to express my gratitude for allowing me to evaluate the manuscript entitled on ‘Exotic and Invasive Plant Species Study and Mapping in Gullele Botanic Garden, Addis Ababa, Ethiopia’. I was enjoying in reading of the manuscript and come up with the following general and spesfic comments:

General comments

i.      The manuscript is highly intriguing and contains some unique thoughts. Due to the lack of adequate scientific data on this issue, the study has its contribution for the scientific community in general. The overall techniques have addressed the objectives and provided evidence through results and discussions.

ii.    Based on my experience in the country in the area of this study, invasive species pose a serious threat to the biodiversity of all plants and could wipe out all plant life. Furthermore, even though the introduction of commercially and environmentally significant non-invasive plants may be significant, their potential impact and invasive status should be investigated. Similarly, the locations of these species should be recorded for track monitoring and follow-up in the future.

iii.  The write-up: The English is generally understandable but some long sentences need thorough revision. On the other hand, the specific name of species should need careful revision (their italics, authors name, repetition) should be checked and revised.

Specific comments and questions for the authors

i.          In the manuscript it states some species were invasive others were potential invasive. Could you elaborate on it (may not be needed in the text)?

ii.        Do you think that 80 sample plots were representative of the 705 hectares of the garden?

iii.      Why did you include the adjacent areas including the roadside in your study? Do you have specific objectives for it?

iv.      Do all the mentioned exotic species were invasive/threats to the native plant species?

Finally, all the specific comments, and questions that require correction, edition and elaboration were included in the manuscript PDF and the authors can find them. 

Comments on the Quality of English Language

The overall English language write-up is understandable. However, it needs a very minor revision. 

Author Response

Dear reviewer 1;

My deepest thanks go out to you for your insightful feedback, which enabled us to improve the manuscript as it ought to be. We have amended the manuscript and come up with point by point responses for the given comments. 

General comments

  1. The manuscript is highly intriguing and contains some unique thoughts. Due to the lack of adequate scientific data on this issue, the study has its contribution for the scientific community in general. The overall techniques have addressed the objectives and provided evidence through results and discussions.

Response: Thank you

  1. Based on my experience in the country in the area of this study, invasive species pose a serious threat to the biodiversity of all plants and could wipe out all plant life. Furthermore, even though the introduction of commercially and environmentally significant non-invasive plants may be significant, their potential impact and invasive status should be investigated. Similarly, the locations of these species should be recorded for track monitoring and follow-up in the future.

Response: Thank you.

iii. The write-up: The English is generally understandable but some long sentences need thorough revision. On the other hand, the specific name of species should need careful revision (their italics, authors name, and repetition) should be checked and revised.

Response: We have attempted to revise the English language of the whole document

Specific comments and questions for the authors

  1. In the manuscript it states some species were invasive others were potential invasive. Could you elaborate on it (may not be needed in the text)?

Response: The difference between potentially invasive species and invasive alien species are explicitly described in the introduction part from line number 45-70. As it indicated in the line number 45 – 53, studies conducted by  Pyšek et al. (2004) and  Kueffer et al. (2004) revealed that the potentially invasive species are species that can lead to problems when they spread to areas where they are not wanted, potentially causing invasive species issues. Whereas the cases of invasive alien species are defined by Kolar and Lodge (2001), that they are non-indigenous species that spreads from the point of introduction and become abundant from the normal range of native plant population (see line number 54 -60).

  1. Do you think that 80 sample plots were representative of the 705 hectares of the garden?

We have gathered 80 plant species from 64 sample plots (not from 80 from sample plots as indicated by the reviewer). According to our species accumulation curve test, the graph has been declining while the number of plots tends to the >60th-64th sample points. This suggests that the probability of finding new species after 64 sample point will be either it is negligible number or no new species will be found. Hence, the sample size take for this study is sufficient. However, by considering the threats of the invasive, species every species should be explored in each pockets of the study area. Therefore, this study recommends conducting the invasive species explicitly and come up with their impacts and management practices. Currently, the impact and management practices study of invasive species are underway. 

iii.      Why did you include the adjacent areas including the roadside in your study? Do you have specific objectives for it?

Response: This is due to the fact that the exotic and invasive species might saturate at the road edge following accessibility by human, animals and vehicles. Accordingly, high numbers of samples were taken from this land use type than other land use types. The research also supports this hypothesis.

  1. Do all the mentioned exotic species were invasive/threats to the native plant species?

Response: Not all exotic species have adverse impacts on the native plants, however, the invasive species have. In this study these exotic species that affect slightly or moderately for the growth and distribution to the native species are categorized as ‘potential invasive species’, whereas these species that has national or regionally designated as invasive alien species categorized as ‘invasives’

Response for comments on the PDF

  1. Comments in Line #30: the highlighted figure is the total density rather than cover abundance. Cover abundance is a visual tool used in the field to identify and quantify species in a sample plot. It is subsequently enlarged from 0 to 100% to generate the (1–9) Van der Maarel (1979) ordinal scales.
  2. Comments on line 41: the overlapping of sentences has been revised.
  3. Comments in Line #100: regarding the ‘level of invasiveness’; revised by the potential of invasiveness
  4. Comments in Line #291-292: request which habit of the species is dominant; the herbaceous species were dominant followed by shrubs and trees. Accumulatively, woody species (tree + Shrub) were dominant than herbs. Accordingly, we have revised manuscript.

Finally, all the specific comments, and questions that require correction, edition and elaboration were included in the manuscript PDF and the authors can find them. 

NB: The comments attached in the PDF has been incorporated and revised accordingly.

Reviewer 2 Report

Comments and Suggestions for Authors

Dear Editor,

The present manuscript, entitled “The study of Exotic and Invasive plant species in Gullele Botanic Garden, Addis Ababa, Ethiopia” presents a comprehensive study of exotic and invasive plant species in Gullele Botanic Garden and compares the diversity of the different habitat types. 

I would like to recommend this manuscript for possible publication in the Journal of Zoological and Botanical Gardens after a minor revision. I found several minor mistakes in the manuscript which are highlighted in the attached pdf file.

These highlighted comments and suggestions must be addressed in detail, and further necessary details (regarding the author name of the species in different tables that should be provided) should be added to the finalized version before formal acceptance.

Regards

Author Response

Dear Reviewer 2!

Thanks for your insightful feedbacks and suggestions We have genuinely gone through your thoughtful comments and revised accordingly. Some comments are seen point by point and described here below:

  1. Comment on the title: is that Botanic or botanical? The institution is designated Gullele Botanic garden, thus Botanic is the right one. For further information: https://gullelebotanicgarden.yolasite.com/more-info.php

  1. Comments in the line #20; was corrected by ‘The ArcGIS was used to track the geographical distribution and map the exotic and invasive/and potential invasive species existed in all land use types of the garden’

  1. Comments in Line #197; it is habit of species rather than habitat.

Finally, the remaining comments have been taken and the manuscript has been revised accordingly.

Reviewer 3 Report

Comments and Suggestions for Authors

The manuscript presents a study on the identification and mapping of non-native plant species in the Gullele Botanic Garden, Addis Ababa, Ethiopia. It focuses on the impact of invasive alien plant species on native and indigenous plant communities within the garden. The study employs vegetation data analysis, floristic similarity analysis, and geospatial data recording to assess the distribution and potential impact of exotic and invasive plant species. The findings highlight the threat posed by these species and emphasize the need for management techniques to prevent their spreading.

1. The manuscript lacks a detailed description of the methodology used for the identification and mapping of non-native plant species. The specific techniques and protocols for data collection, including the sampling methods and data analysis procedures, need to be clearly outlined to ensure reproducibility and reliability of the study.

2. The data analysis section lacks clarity and detail, particularly regarding the computation of diversity indices and the floristic similarity analysis. The manuscript should provide a more comprehensive explanation of the statistical methods used and the rationale behind their selection to ensure the robustness of the findings.

3. The discussion section does not sufficiently address the ecological and conservation implications of the findings. It should elaborate on the potential ecological impacts of the identified invasive species and provide recommendations for effective management strategies to mitigate their effects on native plant communities. Also claims such as "their special distribution is too high and anticipated to have a high potential of dominating the native plant species" was not substantiated throughout the manuscript.

Comments on the Quality of English Language

 Some sections of the manuscript exhibit language issues and lack clarity in conveying the intended meaning. For instance, in

line 39 "According to [25] there are some exotic species which needs causes or cultivation to introduce to need territory or some may become naturalized and those sustain self-replacing populations“ is confusing.

line 54 "Various definitions are given for the term invasive alien species define invasive alien species as non-indigenous species that spreads from the point of introduction and become abundant from the normal range of native plant population." contains grammatical errors and should be rewritten for clarity.

line 78"This most probably associated with the majority restoration and planting works in Ethiopia are executed without considering the source of the materials, even in diverse forest regions, national parks, and damaged ecosystems." contains grammatical errors and can be rewritten as "This is most likely because the majority of restoration and planting works in Ethiopia are executed without considering the source of the materials, even in diverse forest regions, national parks, and damaged ecosystems."

line 82 "Botanic gardens in Ethiopia are recently being practised to conserve indigenous, endangered, endemic and economically important plant species." is awkward and can be rewritten as "Botanic gardens in Ethiopia are recently being utilized to conserve indigenous, endangered, endemic and economically important plant species."

line 159 "The data relating to the abundance, frequency and density of exotic species was described and analyzed XL spreadsheet." contains grammatical errors and can be rewritten as "The data relating to the abundance, frequency and density of exotic species was described and analyzed using Excel spreadsheet."

line 289 "The introduction of these species to the  garden might be either deliberately by planting for their beneficial qualities or unintentionally through imported seeds, vehicles or other vectors and pathways." contains grammatical errors and can be rewritten as "The introduction of these species to the  garden might occur either deliberately by planting for their beneficial qualities or unintentionally through imported seeds, vehicles or other vectors and pathways."

Author Response

Dear Reviewer 3!

My deepest thanks go out to you for your insightful feedback, which enable us to improve the manuscript as it ought to be. We have amended the manuscript and come up with point-by-point responses for the given comments. 

The manuscript presents a study on the identification and mapping of non-native plant species in the Gullele Botanic Garden, Addis Ababa, Ethiopia. It focuses on the impact of invasive alien plant species on native and indigenous plant communities within the garden. The study employs vegetation data analysis, floristic similarity analysis, and geospatial data recording to assess the distribution and potential impact of exotic and invasive plant species. The findings highlight the threat posed by these species and emphasize the need for management techniques to prevent their spreading.

  1. The manuscript lacks a detailed description of the methodology used for the identification and mapping of non-native plant species. The specific techniques and protocols for data collection, including the sampling methods and data analysis procedures, need to be clearly outlined to ensure reproducibility and reliability of the study.

Response: we have made a certain revisions as per the indicated comments.

  1. The data analysis section lacks clarity and detail, particularly regarding the computation of diversity indices and the floristic similarity analysis. The manuscript should provide a more comprehensive explanation of the statistical methods used and the rationale behind their selection to ensure the robustness of the findings.

Response: Sorensen’s similarity coefficient was used to determine beta-diversity or the pattern of species turnover of species between two adjacent ecosystems/habitats/land use types or communities obtained by comparing the number of species and followed the Kent (2012) In this case the floristic comparison was between the sampled land use types. We made a sort of amendment according to this context.

  1. The discussion section does not sufficiently address the ecological and conservation implications of the findings. It should elaborate on the potential ecological impacts of the identified invasive species and provide recommendations for effective management strategies to mitigate their effects on native plant communities. Also claims such as "their special distribution is too high and anticipated to have a high potential of dominating the native plant species" was not substantiated throughout the manuscript.

Response: The sentences "their spatial distribution is too high and anticipated to have a high potential of dominating the native plant species" has been removed. Some points on the potential impacts of the invasive species and way outs on how to manage has been added according to your comment.

Comments on the Quality of English Language

Some sections of the manuscript exhibit language issues and lack clarity in conveying the intended meaning. For instance, in line 39 "According to [25] there are some exotic species which needs causes or cultivation to introduce to need territory or some may become naturalized and those sustain self-replacing populations“ is confusing.

line 54 "Various definitions are given for the term invasive alien species define invasive alien species as non-indigenous species that spreads from the point of introduction and become abundant from the normal range of native plant population." contains grammatical errors and should be rewritten for clarity.

line 78"This most probably associated with the majority restoration and planting works in Ethiopia are executed without considering the source of the materials, even in diverse forest regions, national parks, and damaged ecosystems." contains grammatical errors and can be rewritten as "This is most likely because the majority of restoration and planting works in Ethiopia are executed without considering the source of the materials, even in diverse forest regions, national parks, and damaged ecosystems."

line 82 "Botanic gardens in Ethiopia are recently being practised to conserve indigenous, endangered, endemic and economically important plant species." is awkward and can be rewritten as "Botanic gardens in Ethiopia are recently being utilized to conserve indigenous, endangered, endemic and economically important plant species."

line 159 "The data relating to the abundance, frequency and density of exotic species was described and analyzed XL spreadsheet." contains grammatical errors and can be rewritten as "The data relating to the abundance, frequency and density of exotic species was described and analyzed using Excel spreadsheet."

line 289 "The introduction of these species to the  garden might be either deliberately by planting for their beneficial qualities or unintentionally through imported seeds, vehicles or other vectors and pathways." contains grammatical errors and can be rewritten as "The introduction of these species to the  garden might occur either deliberately by planting for their beneficial qualities or unintentionally through imported seeds, vehicles or other vectors and pathways."

Response: an attempt has been made to improve the English language of the entire document as well as the provided specific comment. All suggested correction has been taken directly.

Reviewer 4 Report

Comments and Suggestions for Authors

The article (jzbg-2768758) presented a study of 'Exotic and Invasive plant species in Gullele Botanic Garden, Addis Ababa, Ethiopia'. Although their findings provide valuable information for managing and conserving both native and exotic plant species in a specific garden in Ethiopia but still the manuscript need revisions before publication. For example; (i) On what basis authors have selected the study areas, briefly describe in M&M, (ii) Discussion section need extensive revision and improvement by comparing your findings with latest studies, (iii) References cited are not sufficient (specially discussion section) to justify/ warrant its publication, (iv) Conclusion is not appropriate. Thus I recommend its acceptance after major revisions.       

Comments on the Quality of English Language

Minor editing of English language required

Author Response

Dear reviewer 4!

I sincerely appreciate your thoughtful comments, which helped us to improve the manuscript. We have amended the manuscript and come up with point by pint responses for the given comments. 

The article (jzbg-2768758) presented a study of 'Exotic and Invasive plant species in Gullele Botanic Garden, Addis Ababa, Ethiopia'. Although their findings provide valuable information for managing and conserving both native and exotic plant species in a specific garden in Ethiopia but still the manuscript need revisions before publication.

For example; (i): on what basis authors have selected the study areas, briefly describe in M&M

Response: We tried to explain the basis to conduct the study in the eventual paragraph of the introduction. This is due to the objective of the study garden is to conserve iindigenous, threatened, endemic, endangered, economically important plants and rare species found in the country. However, we’ve seen different exotic as well as invasive/potentially invasive species in the garden, which contradicts with the organization’s objective. Thus, we suggest that conducting the types, distribution, abundance and the potential of invasiveness of these exotic plant species inside the Botanic garden should be study. We have also address these issues in the objective and method and supported by result and discussions.

(ii) Discussion section needs extensive revision and improvement by comparing your findings with latest studies

Response: we have done some revisions accordingly.

 (iii) References cited are not sufficient (specially discussion section) to justify/ warrant its publication

Response: we appreciate this suggestion; however, studies on this topic as well as in such garden are too rare.  Nevertheless, we made some modification in accordance the suggestions.

(iv) Conclusion is not appropriate. Thus I recommend its acceptance after major revisions.

Response: this was also other reviews comment and we did a significant revision.

Round 2

Reviewer 3 Report

Comments and Suggestions for Authors

The authors have addressed previous review comments with specific revisions and clarifications. They amended the methodology section for better clarity and reproducibility, particularly highlighting the techniques and protocols used for data collection and analysis. In response to concerns about the data analysis section, they provided more details on the computation of diversity indices and the use of Sorensen’s similarity coefficient for floristic comparison. For the discussion section, they removed unsupported claims and added information on the potential impacts of invasive species and management strategies. Language issues identified in various sections were also rectified to improve clarity and coherence. Overall, the authors seem to have made a comprehensive effort to address the feedback and enhance the quality of the manuscript.

Reviewer 4 Report

Comments and Suggestions for Authors

Authors have incorporated all the comments and improved the manuscript